# Role of Macrophages and RhoA Pathway in Atherosclerosis

**DOI:** 10.3390/ijms22010216

**Published:** 2020-12-28

**Authors:** Malgorzata Kloc, Ahmed Uosef, Jacek Z. Kubiak, Rafik Mark Ghobrial

**Affiliations:** 1Houston Methodist Research Institute, Houston, TX 77030, USA; aousef@houstonmethodist.org (A.U.); RMGhobrial@houstonmethodist.org (R.M.G.); 2Department of Surgery, Houston Methodist Hospital, Houston, TX 77030, USA; 3M.D. Anderson Cancer Center, Department of Genetics, University of Texas, Houston, TX 77030, USA; 4Department of Regenerative Medicine and Cell Biology, Military Institute of Hygiene and Epidemiology (WIHE), 01-001 Warsaw, Poland; jacek.kubiak@univ-rennes1.fr; 5Cell Cycle Group, Institute of Genetics and Development of Rennes (IGDR), Faculty of Medicine, Univ Rennes, CNRS, UMR 6290, 35000 Rennes, France

**Keywords:** macrophages, foam cells, RhoA, atherosclerosis, inflammation

## Abstract

The development, progression, or stabilization of the atherosclerotic plaque depends on the pro-inflammatory and anti-inflammatory macrophages. The influx of the macrophages and the regulation of macrophage phenotype, inflammatory or anti-inflammatory, are controlled by the small GTPase RhoA and its downstream effectors. Therefore, macrophages and the components of the RhoA pathway are attractive targets for anti-atherosclerotic therapies, which would inhibit macrophage influx and inflammatory phenotype, maintain an anti-inflammatory environment, and promote tissue remodeling and repair. Here, we discuss the recent findings on the role of macrophages and RhoA pathway in the atherosclerotic plaque formation and resolution and the novel therapeutic approaches.

## 1. Introduction

Atherosclerosis is a chronic inflammatory disease driven by the accumulation of lipids in the wall of the artery. The artery wall is built of several distinct layers of cells: 1. Tunica intima (or intima) is the most internal layer. It faces the artery lumen/blood and contains one layer of the tightly interconnected endothelial cells (forming the endothelial barrier) underlined by the layer of the extracellular matrix, which is situated on top of the layer of elastic fibers (internal elastic lamina); 2. tunica media (or media) is the middle layer. It contains the smooth muscle cells (SMCs) surrounded by the extracellular matrix and located on top of the external elastic lamina; and 3. tunica adventitia (or tunica externa, or adventitia) is the most outer layer. It is built of the connective tissue with embedded fibroblasts, myofibroblasts, resident stem cell progenitors, and the quiescent resident immune cells [1,2].

The development of atherosclerosis is a decades-long process, which progresses through six consecutive stages of lesion formation in the wall of the artery. The initial stage I lesions involve the formation of small lipid deposits in the intimal layer, and the stage II lesions involve large lipid deposits and lipid-containing foam cells in the arterial wall just beneath the endothelium, called the fatty streaks. Stage I and II are called early lesions. These lesions are relatively small and local, and although in addition to the lipids, they also contain lipid-laden cells, called the foam cells, they do not affect the extracellular matrix and do not or minimally change the intima, media, or adventitia architecture. Such early lesions often occur in young people, and they either remain stable, regress, or progress to more advanced lesions. In the stage III (late or advanced) lesions, the accumulation of lipids, foam cells, and immune cells leads to the disorganization and thickening of the intima, media, and adventitia and the proceeding deformation of the artery wall [3]. Advanced atherosclerotic lesions can be further divided into IV, V, and VI subtypes.

The type IV lesions (the atheroma) consist of the extensive region of lipid accumulation (the lipid core) in the intima. The lipid core causes severe disorganization of the intima. Type V lesions are characterized by the formation of fibrous connective tissue. When the fibrotic tissue forms on top of the lipid core, these types of lesions are called type Va or fibroatheroma. When the lipid core and the surrounding tissues are calcified, the lesion is classified as type Vb or calcific lesions (or sometimes called type VII lesions). The type V lesions lacking the lipid core but very rich in fibrotic tissue are designated as type Vc or fibrotic lesions (or sometimes also called type VII lesions). The type V lesions cause the narrowing of the artery lumen. If type IV or V lesions develop surface disruption, hematoma, hemorrhage, or thrombotic deposits, which are the main cause of atherosclerosis morbidity and mortality, they are classified as type VI or complicated lesions. Type VI lesions are further subdivided into type VIa (with a disrupted surface), type VIb (with hematoma or hemorrhage), type VIc (with thrombosis), and type VIabc (combining all these features) [4]. Some of these lesions can progress to necrosis and necrotic disintegration, which leads to luminal thrombosis followed by the ischemia of the heart muscle, heart attack, stroke, or even cardiac death (Figure 1) [5].

Below, we describe why and how the lipids accumulate in the arterial wall, what are the roles of macrophages in lipid accumulation and the development and progression of atherosclerosis, and how the RhoA pathway regulates some of these processes.

## 2. Aggregation of Lipids in the Arterial Wall

Low-density lipoproteins (LDLs) are the main carriers of circulating cholesterol and one of the risk factors for the development of atherosclerosis. Plasma LDLs particles contain cholesterol esters surrounded by a monolayer of phospholipids, free cholesterol, and one molecule of apolipoprotein B (apoB-LPs) [6]. The LDLs are the main source of lipid droplets accumulating in the arterial wall. There are two main theories on the reason and mechanism of lipid accumulation: 1. The response-to-injury theory, and 2. the response-to-retention theory [6,7,8,9].

*The response-to-injury theory* states that the atherogenesis starts with a very subtle and focal physical or chemical injury to the vessel endothelium, caused by physical trauma, high blood pressure, turbulent blood flow, high glucose or lipids level, free radicals, or various inherited metabolic defects. This is followed by the adherence of platelets, which secrete the platelet-derived growth factor (PDGF) promoting the proliferation of SMCs. When this theory was conceived in the 1970s by Russel Ross, it was believed that the over-proliferation of SMCs, resulted over time, in artery occlusion and that the SMCs were the major factor in the formation and progression of the atherosclerotic lesions. However, later studies showed that the proliferation of SMCs in the plaque is partially beneficial, due to the fact that it stabilizes the plaque [10]. There is also a controversy on the subject of how the arterial endothelium allows monocytes entry to the arterial wall. One proposition assumes that the injured endothelial cells change shape, loosen the intercellular tight junction, thus increasing the permeability of the endothelial barrier, and producing cytokines, which recruit the immune cells. They also produce adhesion molecules such as vascular cell adhesion molecule 1 (VCAM-1), which allows monocytes and T-cells to attach to the endothelial surface. The monocytes and T-cells squeeze between the loosened endothelial junctions and enter the intima, where the monocytes differentiate into macrophages [11]. However, the early studies proposed that the initial entry of the monocytes proceeded through the intact endothelium [12]. Once in the arterial wall, the immune cells induce the production of cytokines and the inflammatory cascade within the arterial wall [10]. There are over 50 members of the cytokine superfamily. There are several classes of cytokines: Interleukins (ILs), interferons (IFNs), tumor necrosis factors (TNFs), transforming growth factors (TGFs), colony-stimulating factors (CSFs), and chemokines (specific cytokines adapted for chemotaxis) [10]. Based on the structure of their receptors, the cytokines are also divided into class I and II, which function through the JAK-STAT (Janus kinase-signal transducers and activators of transcription) pathway and include the majority of ILs, CSFs, and IFNs. In contrast, the IL-1 family containing IL-1α, IL-1β, IL-18 and Il-1ra, and the TNF family function through the mitogen-activated protein (MAP) kinase and the nuclear factor-κB (NF-κB) pathways, while the TGF-β superfamily, containing TGF-β1–2-3, activins, inhibins, myostatin, anti-Mullerian hormone (AMH), and bone morphogenetic proteins (BMPs), act through the cell-surface serine/threonine kinase receptors of the intracellular mediators Smads [10,13]. Another classification of cytokines depends on their pro-inflammatory or anti-inflammatory functions. For example, the TNF, IL-1, IL-12, IL-18, IFN-γ are pro-inflammatory and IL-4, IL-10, IL-13, TGF-β are anti-inflammatory. In the context of atherosclerosis, Tedgui and Mallat [10] proposed another classification of cytokines as the pro-atherogenic (tumor necrosis factor, TNFR, and IL-1 families, class I and II cytokines, etc.) or anti-atherogenic (IL-1ra, IL-18BP, IL-6, IL-9, IL-10, and TGF-β), for the comprehensive lists of factors, see Tables in Tedgui and Mallat [10]. Leaving their function aside, all cytokines share certain commonalities: They act as paracrine, endocrine, or juxtacrine factors, their effects are tissue-, time, and context-specific, they function in synergy with other cytokines, which amplifies their effect, and they often share the same subunit of the receptors recognizing the particular cytokine family [10].

Studies showed that in the mouse atherosclerosis model, blocking the chemokines and/or their receptors decreases inflammation and inhibits or slows down the progression of atherosclerosis [14]. Due to the leaky endothelial barrier not only the immune cells but the lipoproteins, such as LDLs, enter the wall of the artery, where, after the exposure to macrophages, nitric oxide, and lipoxygenase enzymes, they undergo oxidation. Oxidized LDLs are engulfed by the macrophages, which transform into lipid-laden foam cells. These lipid-laden macrophages are unable to migrate efficiently and resolve the inflammation, which contributes to the progression of the atherosclerotic lesions. Although many of the foam cells will enter apoptosis and die, the lipids will remain in the vessel wall. Additionally, over time, the recruited intima T-cells produce cytokines that induce the migration of the smooth muscle cells from the media to the intima. In response to the growth factors, the smooth muscle cells proliferate. Some of the smooth muscle cells and dendritic cells can also accumulate lipids and transform into lipid-laden foam cells [15]. The multiplied SMCs, foam cells, and accumulated lipids cause the uplifting of the endothelial layer and decrease of the arterial lumen [11,16,17,18].

The second theory explaining the aggregation of lipids in the arterial wall is the *response-to-retention theory*. This theory states that the circulating lipoprotein particles are entrapped in the subendothelial space, and subsequently become bound and retained by the components (for example, the proteoglycans) of the subendothelial extracellular matrix of the arterial wall. The hydrolytic and oxidative enzymes present in the extracellular matrix induce oxidation, proteolysis, and lipolysis of LDLs. Such modified LDLs, coalesce, fuse, and aggregate which prevents them from leaving the artery wall [6,19,20]. Studies in rabbits showed such entrapment and accumulation after the injection of large quantities of human LDLs into the circulation [21,22]. The aggregates of fused lipids induce, probably by the upregulation of the LDL receptors, the formation of macrophage- and smooth muscle cell-derived foam cells [6,23]. As in the described above response-to-injury theory, the accumulation of the LDLs leads to the recruitment of the immune cells, and the cascade of inflammatory responses, which leads to atherosclerosis. The most probable is the combination of these two theories, and that the very subtle physical or chemical injuries to the vessel endothelial layer allow the initial entry of the lipids into the vessel wall. A very important factor in the development and progression of atherosclerosis is how the endothelial cells, which are in constant contact with LDLs, metabolize, and process the LDLs. Recent studies showed that when the endothelial cells take up too much LDL, they form the cholesterol crystals, which are deposited on the basolateral membrane. This makes the endothelium inflamed and leaky and disrupts the normal functioning of the endothelial barrier, allowing free entry of cholesterol and monocyte infiltration. The cholesterol crystals can also form in the macrophages by active intracellular processing of cholesterol, which activates inflammasomes and progresses atherosclerotic plaques [24,25].

Another, often neglected, drivers of lipid accumulation and foam cell formation in atherosclerosis are the platelets (Figure 2 and Figure 3).

It has been recently recognized that the platelets not only play a role in thrombosis but also modulate inflammation, cause endothelial dysfunction, and initiate the formation of atheromatous plaques. Studies showed that platelets adhere to the activated, by the inflammatory events (such as a mechanical injury), endothelial cells of the vessel wall. The adhesion of platelets to endothelium induces the recruitment of monocytes and their subsequent differentiation into macrophages and the transformation into foam cells (Figure 2 and Figure 3) [26].

Macrophages have a critical role in the development of atherosclerotic plaques. Their precursors (monocytes) recruited into the sub-endothelial space differentiate into the pro-inflammatory M1 macrophages. These macrophages secrete pro-inflammatory cytokines, which recruit more monocytes to the plaque and promote their differentiation into M1 macrophages. The adhesion of the monocytes to the endothelium is controlled by the ligands of C-C chemokine receptor type-1 (CCR1 or CD191), C-C chemokine receptor type 5 (CCR5 or CD195), and C-X-C chemokine receptor type 2 (CXCR2), such as chemokine (C-C motif) ligand 5 (CCL5 or RANTES), CCL3, CCL4, and the C-X3-C Motif Chemokine Ligand 1 to 7 (CX3CL1 to CX3CL7), respectively. The survival and activation of macrophages and the formation of the foam cells within the plaque are controlled by CXCL5, CXCL4, and CX3CL1 [25]. Once the macrophage capacity to neutralize (esterify) the toxic free cholesterol has excided, the engulfment of lipids will result in macrophage death, followed by the necrotic processes within the plaque, its destabilization, and rupture (Figure 2 and Figure 3) [15]. However, there are also the anti-inflammatory M2 macrophages that when present within the atherosclerotic plaque can resolve inflammation and diminish or reverse plaque formation. Studies in the atherogenic mice models showed that the activation of the peroxisome proliferator-activated receptor-gamma (PPAR-γ) pathway or treatment with IL-13 induces M2 macrophage polarization, increases the plaque collagen content, and inhibits the progress of atherosclerosis [27,28]. Since the macrophages can switch, depending on the milieu, between M1 and M2 phenotypes, the manipulation of the plaque macrophage phenotype may be a viable option for the prevention or reversal of atherosclerosis in humans.

It is also important to signal here the beneficial role of high-density lipoprotein (HDL) cholesterol in atherosclerosis. Among the lipoproteins, the HDL that contains the highest proportion of the proteins is the densest [29]. It has been shown that the HDL protects against the initiation and progression of atherosclerosis [30]. One of the many protective mechanisms of HDL relies on the absorption of cholesterol from the blood and facilitating its transport to the liver with a subsequent removal through the gallbladder and the biliary ducts. Another mechanism is the protection against the formation, or even causing the regression of the foam cells. The HDL protein component—the apoA-I interact with the macrophages and/or foam cells, and through the ATP binding cassette (ABC) transporters ABCA1 and ABCG1, which are membrane lipid translocases, facilitate the efflux of excess cholesterol [31]. Additionally, the apoAI neutralizes ROS, which promote inflammation and oxidative modification of LDL, and accelerate atherosclerosis [24].

Below, we describe the origin, formation, and function of foam cells in atherosclerosis (Figure 2 and Figure 3).

## 3. The Foam Cells

The foam cell is the name given to any cell-laden with lipids. Although a variety of cell types, such as smooth muscle cells, epithelial cells, endothelial cells, mesangial cells, and liver Kupffer cells, can transform into foam cells, it has been always believed that the monocyte/macrophage-derived foam cells are the major factors in the development and progression of the atherosclerotic plaque [15,32,33,34,35,36,37,38]. However, the recent studies indicate that between 60–70% of foam cells in mouse atherosclerotic lesions and >50% in human lesions have the non-leukocyte (mainly smooth muscle cells) origin [33,34]. In the healthy blood vessels, the smooth muscle cells provide contractility and stabilize the vessel wall by polymerizing the soluble type I collagen into insoluble fibers. Studies showed that the lipid-laden smooth muscle cells are unable to form collagen fibers, which may, further down the road, lead to plaque vulnerability [15]. This indicates that our knowledge of the origin of the foam cells in atherosclerosis is very far from being complete and adamant. Additionally, recent studies indicate that the foam cell precursor cell type, the mechanism of foam cell formation, the type of lipids they accumulate, and the functions they perform are very disease-specific [35]. For example, the myelin-laden foam cells in the brain lesions of the multiple sclerosis patients are associated with the demyelinating activity, and in tuberculosis, the macrophage-derived foam cells are mainly laden with triglycerides, and not cholesterol derivatives such as the atherosclerotic foam cells [35,36,37].

The first step in the formation of foam cells is the engulfment of the oxidized and acetylated low-density lipoproteins (oxLDLs and AcLDLs), and minimally modified LDLs (mmLDLs) by the monocyte/macrophages, which were recruited into the vessel wall. Recent studies indicate that some of the foam cells may derive not from the recruited but from the subpopulation of resident macrophages present in the arterial intima [38]. The LDLs are internalized by macropinocytosis, phagocytosis, but mainly through the binding to the scavenger receptors (SRs) present on the macrophage surface, such as, for example, cluster of differentiation 36 (CD36 also called the scavenger receptor class B member 3 (SCARB3), and SR-A (also called the CD204) (Figure 2 and Figure 3), [26,39,40,41]. The foam cell formation can be also induced by certain pathogens, such as *Porphyromonas gingivalis* and *Chlamydia pneumoniae*, which induce LDLs uptake by the macrophages [42].

As the development of the atherosclerotic plaque progresses, the foam cells overladen with lipids undergo apoptosis. Under nonchronic inflammatory conditions, the macrophages recognize apoptotic cells through the presence of exposed phosphatidylserine (PtdSer) that in viable cells is confined to the inner, unexposed surface of the cell membrane, and remove them through the process of programmed cell removal (PrCR), or efferocytosis. This prevents inflammation and promotes tissue repair. In the plaque, the efferocytosis is impaired, and the inefficiently cleared dead foam cells, undergo necrosis, which induces an additional inflammatory response. Formation of the necrotic core within the plaque increases its vulnerability to rupture. Studies show that the impairment of efferocytosis in the plaque is caused by a lower expression of the calreticulin (the so-called “eat me” ligand) which promotes the engulfment of dead cells, and the phagocytotic inefficiency of lipid exposed macrophages and SMCs, which also have a great efferocytosis capacity [43,44]. Under normal conditions the dead cells are removed within minutes, while the efferocytosis in the plaque is reduced nearly 20-fold [43,44]. Mouse and human studies showed that the development of necrotic plaque with a thin (vulnerable to breakage) collagen cap is regulated by the macrophage Ca^2+^/calmodulin-dependent protein kinase γ (CaMKIIγ). Advanced and necrotic plaques show activation of CaMKII, and the macrophage-specific deletion of CaMKII results in the decreasing of necrotic cores and thickening of the collagen caps, which increase the stability of the plaque [45]. Additionally, the CaMKIIγ-deficient macrophages upregulated the expression of the activating transcription factor 6 (ATF6) that induces liver X receptor-α (LXRα) expression, which in turn regulates macrophage efferocytosis [45].

### Foam Cell Activities

Many in vitro and rodent model studies showed that the macrophage-derived foam cells have inflammatory functions. The internalized oxLDLs induce, through the binding to the scavenger receptor/Toll-like receptors (CD36/TLR4/TLR6) complex, transcription and translation of the pro-inflammatory cytokines. Moreover, the cholesterol and the lipids isolated from human atheromas activated NOD-, LRR-, and pyrin domain-containing protein 3 (NLRP3) inflammasome, induced endoplasmic reticulum (ER) stress, and the production of pro-inflammatory IL-1β in the in vitro cultured macrophages [35,46,47,48]. However, there are also data showing that lipid-laden macrophages have a lower expression of inflammatory genes encoding cachectin (TNFα), IL-1β, cyclooxygenase-2 (COX-2), neutrophil recruitment cytokine CXCL8, T-cell activation and inflammatory responses inducing CCL19, and suppressed activation of NF-κB and secretion of TNFα [35].

Animal and human studies showed that foam cells produce matrix degrading metalloproteinase (MMPs) [35]. The foam macrophages produce more MMP-14 and less MMP-3 inhibitor (TIMP-3) than normal macrophages. Such excessive degradation of the extracellular matrix by the MMPs may lead to the destabilization of the atherosclerotic plaque and its rupture [35]. There are also studies on the tuberculosis models indicating that the foam macrophages have a very limited anti-microbial activity; showing impaired phagocytosis, maturation of phagosomes, and respiratory burst. Additionally, as a side effect of lipid accumulation, the foam cells become an unintended nutritional source for the pathogens, which often after ingesting the macrophage-derived lipids become dormant and more resistant to the antibacterial drugs [35,49].

There are also other, recently discovered factors, which regulate lipid uptake, functions of macrophages, and the progression and stability of the atherosclerotic plaque. The RNA sequencing of stable and unstable atherosclerotic plaques identified 47 differentially expressed long non-coding RNAs (lncRNAs) [50]. It is known that only a small fraction of the mammalian genome is transcribed into the protein-coding mRNAs, while about 80% of the genome is transcribed into non-coding RNAs. Some of these RNAs, especially the long non-coding RNAs (lncRNAs) are important regulators of transcription, chromatin condensation status, and nuclear architecture. In addition to their nuclear functions, they also regulate cytoplasmic processes such as translation, post-translational modifications, and mRNA stability [51]. Studies showed that the expression of one of these RNAs, the LINC01272 (also called PELATON), is highly upregulated in the nuclei of monocytes and macrophages upregulated in the unstable plaques, especially in the regions of plaque inflammation. The knockdown studies showed that although the PELATON RNA is not translated into protein it regulates phagocytosis, lipid uptake, and production of the reactive oxygen species [50]. These findings open new avenues for the studies of factors regulating macrophage functions in atherosclerosis.

## 4. RhoA Pathway Involvement in Atherosclerosis

The small GTPase RhoA, its downstream effectors the protein serine/threonine ROCK kinases and interacting mTORC2 and Rac1 pathways, are the major regulators of actin polymerization and actin-dependent processes in all eukaryotic cells. Therefore, they also regulate macrophage phagocytosis, endo and exocytosis, receptor surface expression and recycling, and macrophage motility (Figure 4) [52,53,54,55,56,57,58,59,60,61].

The studies from our laboratory on the rodent transplantation model showed that macrophage-specific deletion of RhoA or inhibition of ROCK kinase impairs actin-dependent functions of macrophages and prevents them from moving into and destroying the transplanted organs [52,53]. Such RhoA inhibition-based therapy may be potentially applicable for the inhibition of deleterious functions of macrophages in the atherosclerosis. Below, we describe how the RhoA pathway affects macrophage functions, and in general, the development and progression of the atherosclerotic plaques.

The RhoA/ROCK pathway has different and, often, depending on the cellular context, contradictory effects on the cardiovascular system and atherosclerosis. The RhoA regulates cardiac development during embryogenesis and also regulates endothelial cells and VSMCs, promotes angiogenesis, and stabilizes vessels during vascular development. However, it also accelerates pathological changes in the cardiovascular system especially in diabetic and hypertensive patients [62] and affects macrophage behavior in the atherosclerotic plaques. Studies indicate that the cholesterol-laden macrophages have lower levels of GTP-bound active RhoA, lower phosphorylation of myosin light chain, and decreased motility. The expression of constitutively active RhoA prevents cholesterol-dependent inhibition of macrophage motility. These results indicating that the increase in cholesterol content lowers RhoA activation and inhibits macrophage migration may explain why the foam cells become trapped within the atherosclerotic plaques [63].

The members of the Rho family of small GTPases such as RhoA, Rac, and Cdc42 contain a carboxyl-terminal CAAX motif that undergoes posttranslational prenylation (geranylgeranylation and farnesylation) catalyzed by the geranylgeranyltransferase (GGTase-I) enzyme, which makes the carboxy terminus more hydrophobic allowing the attachment and interaction with the cellular membranes [64]. There have been contradictory results on the effect of geranylgeranylation on the activity of RhoA. Some studies indicated that the knockout of macrophage GGTase induced the accumulation of the activated (GTP-bound) RhoA, Rac1, and CDC42, which in turn upregulated the production of proinflammatory cytokines and reactive oxygen species [65], while the other studies showed the opposite effect of decreased activation of RhoA. To solve this controversy, Khan et al. [64] studied the effect of the GGTase-I inactivation in the macrophages on the development of atherosclerosis in the LDL receptor–deficient mice. They found that the lack of GGTase-I decreased the accumulation of the phospholipids and cholesterol esters in the macrophages and increased cholesterol efflux, while the knockdown of RhoA stabilized cholesterol efflux. They concluded that GGTase-I deficiency activates RhoA leading to the increase of reverse cholesterol transport (cholesterol efflux) from the macrophages and the inhibition of atherosclerosis [64]. Additionally, these studies indicate that, at least in the macrophages, the non- geranylgeranylated Rho proteins are active, i.e., GTP-bound, which changes our understanding of the role of CAAC motif prenylation in the macrophages [64].

Other studies of the significance of the RhoA and Rac1 prenylation for atherosclerosis development have focused on the role of the unmodified and geranylgeranylated RhoA and Rac1 in the atherosclerosis progression in the rabbit model [66]. These authors separated from the atherosclerotic lesions, the unmodified and geranylgeranylated forms of RhoA and Rac1, and determined their activity by membrane translocation and a pull-down assay. They showed that in the heritable hyperlipidemic rabbits, which are prone to the myocardial infarction, the level of unmodified RhoA and Rac1 and the levels of a membrane-bound (geranylgeranylated) RhoA and Rac1 increased with age (assessed between 3 and 7 months). The unprocessed forms and total activity of RhoA and Rac1 significantly decreased after 7 months of age (assessed between 15 and 24 months). The authors suggest that these data indicate the importance of the unmodified forms of RhoA and Rac1 and the total RhoA/Rac1 activity for the early stages of atherosclerotic plaque development [66].

The issue of farnesylation and geranylgeranylation of RhoA, Rac1, and CDc42 in atherosclerosis is important for the therapeutic use of the lipid-lowering agents, statins, in the prevention and reduction of atherosclerosis. Independently of the lipid lowering activities statins have a pleiotropic effect, such as prevention of the synthesis of farnesyl pyrophosphate (FPP) and geranylgeranyl pyrophosphate (GGPP), and as a result the prevention of Rho, Rac and Cdc42 modification and attachment to the cell membrane [62]. This, in turn, results in the modulation of the immune response in the atherosclerotic plaques by decreasing the influx of the inflammatory cells and modifying the molecular composition of the plaque lipid core. In vivo and in vitro studies showed that the statins affect the RhoA/ROCK signaling pathways and that RhoA/ROCK inhibitors have a protective effect on the vasculature [67]. Several studies showed that the inhibitor of RhoA/ROCK kinase, Y27632, results in the regression of atherosclerotic plaques in the porcine and mouse model by modulating the T-cell activity [68,69,70].

The RhoA/ROCK activity also affects the function of the vascular endothelium. The increased ROCK activity impairs endothelial cells and the application of ROCK inhibitor, fasudil improves endothelium-dependent dilation of the blood vessels in the patients with the coronary artery disease (CAD), which is characterized by the over-activity of ROCK [67].

Since there is also a definite correlation between hypertension, endothelial damage, and atherosclerosis, some studies suggest that the link between these processes lies in the proinflammatory function of the vasoconstrictor angiotensin II (Ang II) [30,71]. These studies showed that the RhoA Guanine Nucleotide Exchange Factor 1 (Arhgef1) is necessary for Ang II-induced activation of the immune cells. In the mouse model, the deletion of Arhgef1 inhibits Ang II-dependent immune cell activation and their recruitment to the arterial endothelium, which suggests that the Arhgef1 can be a novel target for therapeutic intervention in atherosclerosis [71]. It seems that the beneficial or unfavorable effect of RhoA on atherosclerosis depends on the cellular context. For example, on the one hand, the decreased RhoA activity prevents the immune cells influx into the artery wall and lowers the inflammation, and on the other hand, it traps the macrophages/foam cells and promotes plaque development.

## 5. Summary and Future Perspectives

In conclusion, studies show that the foam cells, both the SMC- and macrophage-origin, play a crucial role in plaque vulnerability to rupture. This implies that they are a very promising target for the improvement of plaque stability. There are also very important new findings on the pleiotropic effects of statins, which in addition to the lipid-lowering activities have a pleiotropic effect including the prevention of Rho, Rac, and Cdc42 modification and their attachment to the cell membrane, which, in turn, decreases the influx of the inflammatory cells to the atherosclerotic plaque. Additionally, the administration of the inhibitors of the RhoA/ROCK pathway regresses the atherosclerotic plaques. On the other hand, the loss of RhoA activity decreases the motility of macrophages/foam cells and causes their entrapment within the plaque. Therefore, due to these contradictory effects of RhoA activity on different aspects and components of the atherosclerotic plaque, the development of novel anti-atherosclerotic therapies, will require further detailed studies of the role of RhoA and collaborating signaling pathways on macrophages and other immune and non-immune cells participating in the development and progression of atherosclerosis. It will also require the identification of the RhoA/ROCK inhibitors approved for clinical use. Some of these novel options may be the Fingolimod or Siponimod, which inhibit the RhoA/ROCK pathway and are already clinically approved for the treatment of multiple sclerosis (MS) [72,73,74,75].

## Figures and Tables

**Figure 1 ijms-22-00216-f001:**
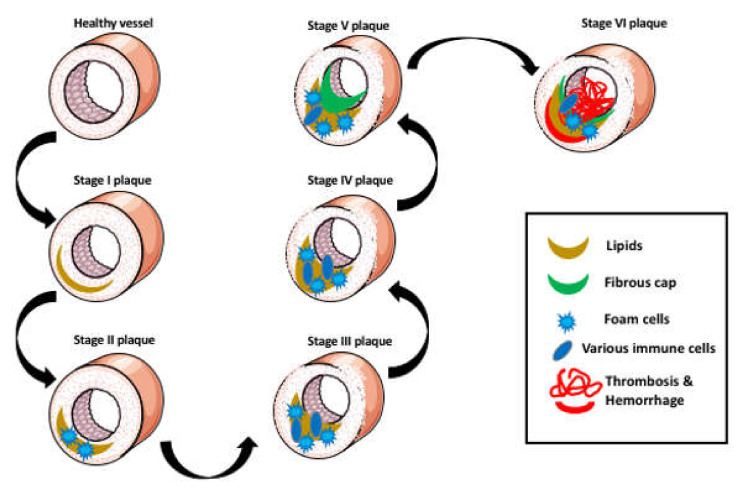
Stages of atherosclerosis progression. The formation of a small area of lipid deposits in the intimal layer is called the stage I lesions. The stage II lesions contain large lipid deposits and lipid-containing foam cells in the arterial wall just beneath the endothelium called the fatty streaks. Stage I and II are called early lesions (plaques). Stage III plaques consist of a large lipid layer filled with the foam cells and other immune cells, which lead to the disorganization and deformation of the artery wall. Stage IV lesions (the atheroma) consist of a large lipid layer. The lipid core causes severe disorganization of the intima. Stage V lesions contain fibrotic tissue on top of the lipid layer. Stage V lesions cause the narrowing of the arterial lumen. Stage VI lesions are vulnerable to plaque rupture, which is followed by thrombosis and hemorrhage.

**Figure 2 ijms-22-00216-f002:**
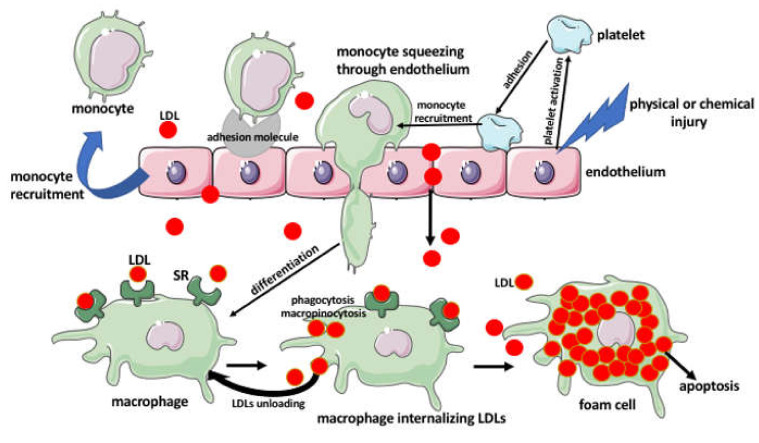
Formation of the macrophage-derived foam cells. The injured endothelial cells change shape, loosen the intercellular tight junction, increasing the permeability of the endothelial barrier, and produce cytokines, which recruit the immune cells. They also produce adhesion molecules such as vascular cell adhesion molecule 1 (VCAM-1), which allows monocytes to adhere and enter the intima. While in the intima, monocytes differentiate into macrophages. Low-density lipoproteins (LDLs) enter the intima from the circulation and become internalized by the macrophages through the binding to the scavenger receptors (SRs) by phagocytosis and macropinocytosis. The macrophage will try to unload the lipids, and if there is a lipid overload the lipid-laden macrophages transform into foam cells that eventually die. Another factor in the recruitment of monocytes to the intima is the platelets. The injured endothelium activates platelets, which adheres to the endothelium and induces further endothelial changes and the monocyte recruitment.

**Figure 3 ijms-22-00216-f003:**
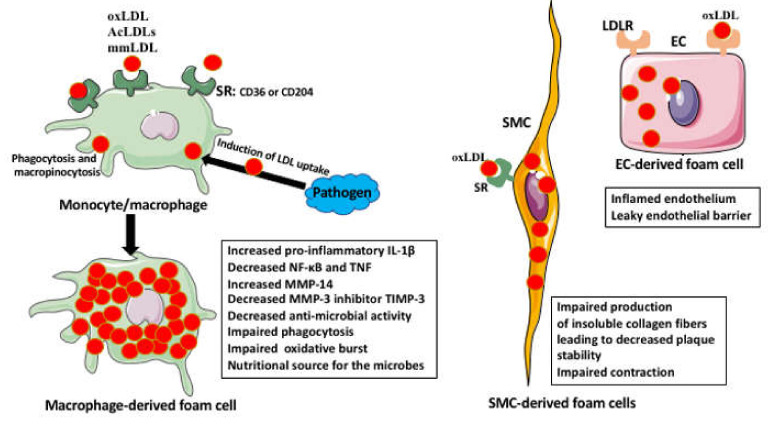
Types of foam cells and effects on cell functions. The macrophage-derived foam cells form from the monocytes/macrophages, which uptake various forms of LDLs (oxLDL, AcLDL, and mmLDL) through the SRs such as CD36 or CD204, phagocytosis or macropinocytosis. The LDL uptake can be also induced by various pathogens. The lipid-laden macrophages (the foam cells) have lowered the production of some of the pro-inflammatory cytokines, impaired phagocytosis, motility, and anti-microbial activity. They also have increased metalloproteinase and decreased metalloproteinase inhibitor production, which can lead to plaque instability. The foam cells can also derive from other cell types such as smooth muscle cells (SMCs) and endothelial cells EC. The SMCs uptake LDL via SR receptors. The lipid-laden SMCs have impaired contraction and impaired production of insoluble collagen fibers, which can lead to decreased plaque stability. The ECs uptake LDLs via various LDL receptors (LDLR). The lipid-laden endothelial cells become inflamed and are unable to properly adhere to each other, which leads to the leakiness of the endothelial barrier. The leaky endothelial barrier becomes permeable for lipids and immune cells.

**Figure 4 ijms-22-00216-f004:**
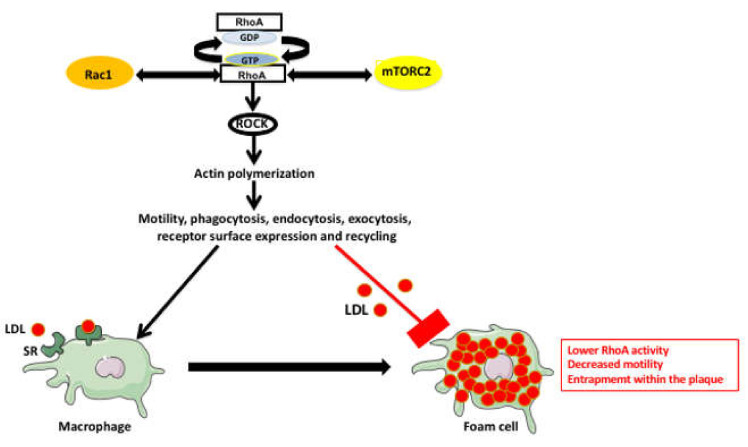
RhoA pathway involvement in macrophage and foam cell functions. The RhoA pathway controls, via ROCK kinase, the actin polymerization and actin-dependent functions in all cells including the macrophages. The RhoA pathway is reciprocally regulated by Rac1 and mTORC2 pathways. The macrophages uptake the LDLs through the SRs and become the foam cells. The foam cells have a lower activity of RhoA, which makes them less motile and leads to their entrapment within the atherosclerotic plaque.

## Data Availability

Not applicable.

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
