# Peer review of "Role of Macrophages and RhoA Pathway in Atherosclerosis"

_ijms, 2020, doi:10.3390/ijms22010216_

Round 1
Reviewer 1 Report
The manuscript by Dr Kloc et al. provides a textbook-like summary for the formation and progression of atherosclerosis while focusing on the roles of macrophages and RhoA therein. In general, the manuscript is not well structured and the novelty of this specific review compared to the literature is not clear. As a review, the manuscript should provide better and more informative illustrations to present the main text, unfortunately I failed to see this. Detailed comments as follow:
Major points
- The title is confusing, suggest to revise, like “Role of macrophages and RhoA pathway in atherosclerosis” or similar.
- Figure 1 does not match well the main text content in terms of disease stages. In the main text, atherosclerosis progression is presented by stages I-VII while in the figure by “vulnerable” and “stable”. This does not help readers to understand the review. The authors, instead, should make their own figure to illustrate the disease progression stages as presented in the main text.
- Same issue for Figure 2. The title is “formation of the foam cells” and the figure illustrates the differentiation of monocytes into macrophage foam cells. This is just an adaption of a classic figure (conventional) without any new components (except the addition of platelets) and too simplified for a scientific review. In fact, the authors did provide some new information regarding the foam cells on page 6 line 197 stating foam cells are not only of macrophage origin but also differentiated from SMCs, ECs etc etc. Similar to Figure 1, the figure does not match the main text.
- RhoA signaling pathway is the other main focus of this manuscript. Given the complex or even controversial findings in the literature, an illustration should be helpful for readers to understand the proposed roles of RhoA pathway in atherosclerosis.
- Structure of the manuscript can be improved. The authors are expected to summarize their findings in literature review and provide future perspectives in the field.
Minor points
- Page 2 line 47, the sentence seems not complete.
- Page 2 paragraph 2, the numbering of stages is not consistent, e.g. type VA vs. type Vb vs. Via vs. VIb etc. Please check through the manuscript.
- Abbreviations should be used only at the first place, please check through (e.g. SMCs)
- Please check through grammars and typos (e.g. plague should be plaque)
- Some citations lack references.
Author Response
Reviewer 1:
Comments and Suggestions for Authors
The manuscript by Dr Kloc et al. provides a textbook-like summary for the formation and progression of atherosclerosis while focusing on the roles of macrophages and RhoA therein. In general, the manuscript is not well structured and the novelty of this specific review compared to the literature is not clear. As a review, the manuscript should provide better and more informative illustrations to present the main text, unfortunately I failed to see this. Detailed comments as follow:
RESPONSE: The manuscript has been extensively revised according to the comments of both reviewers. This changed its structure because of numerous modifications. We hope that the revised version will satisfy the referee.
Major points
- The title is confusing, suggest to revise, like “Role of macrophages and RhoA pathway in atherosclerosis” or similar.
RESPONSE: As suggested we changed the title
- Figure 1 does not match well the main text content in terms of disease stages. In the main text, atherosclerosis progression is presented by stages I-VII while in the figure by “vulnerable” and “stable”. This does not help readers to understand the review. The authors, instead, should make their own figure to illustrate the disease progression stages as presented in the main text.
RESPONSE: As suggested we made a new Figure 1 which follows the stages described in the text
- Same issue for Figure 2. The title is “formation of the foam cells” and the figure illustrates the differentiation of monocytes into macrophage foam cells. This is just an adaption of a classic figure (conventional) without any new components (except the addition of platelets) and too simplified for a scientific review. In fact, the authors did provide some new information regarding the foam cells on page 6 line 197 stating foam cells are not only of macrophage origin but also differentiated from SMCs, ECsetc etc. Similar to Figure 1, the figure does not match the main text.
RESPONSE: We left Figure 2 as it is because we think that it is beneficial for the less informed readers, but as suggested we added an additional figure (new Figure 3) on the origin and types of the foam cells
- RhoA signaling pathway is the other main focus of this manuscript. Given the complex or even controversial findings in the literature, an illustration should be helpful for readers to understand the proposed roles of RhoA pathway in atherosclerosis.
RESPONSE: As suggested we made a new figure (now Figure 4) that illustrates role of the RhoA pathway in foam cells and atherosclerosis. We also added the comments on the controversial role of RhoA pathway in atherosclerosis.
- Structure of the manuscript can be improved. The authors are expected to summarize their findings in literature review and provide future perspectives in the field.
RESPONSE: As suggested we added the summary and the future perspectives and 4 new references.
Minor points
- Page 2 line 47, the sentence seems not complete.
RESPONSE: we corrected the sentence
Page 2 paragraph 2, the numbering of stages is not consistent, e.g. type VA vs. type Vb vs. Via vs. VIb etc. Please check through the manuscript.
RESPONSE: we corrected the numbering of stages
Abbreviations should be used only at the first place, please check through (e.g. SMCs)
RESPONSE: we checked the abbreviations Please check through grammars and typos (e.g. plague should be plaque)
RESPONSE: we corrected the typos and grammar
- Some citations lack references.
RESPONSE: we added references to the citations

Reviewer 2 Report
Kloc M et alter have reviewed the role of macrophages and RhoA pathway in the development of atherosclerosis.
This is a comprehensive review of plaque formation, from the Stary pathological stages to molecular basis.
I believe the authors have performed a great job putting all together, the available information on molecular bases of atherosclerotic plaque formation. I believe the content of the review encompasses the current scientific knowledge and it will be useful for readers.
There are some aspects that I would like to be better addressed:
The role of cholesterol, and particularly LDL-C as a causal factor should be clearly stated. Although it is mentioned across the paper, I feel the importance of cholesterol as endothelium or macrophage activator, or the role of cholesterol crystals inducing inflammasome activation, is somehow diluted among other factors (lines 93-94). I suggest highlighting its etiological role that has been recently reviewed in Boren J et al Eur Heart J, 21;41(24):2313-2330.
I believe that in the context of this review it would be more important to devote some room to the role of HDL than, for example describing all Stary stages. HDL is not mentioned while playing a crucial role on maintaining the artery wall cholesterol burden under control, through its interaction with macrophages.
I suggest including a paragraph on this issue.
It seems that there are two different phenotypes of macrophages in arteriosclerotic plaques. M1 proinflammatory and acting as plaque disruptor and M2, which are anti-inflammatory and taking part of healing mechanisms (Vergallo R et al. NEJM 2020: 383, 9 and Eur Heart J 2020). This aspect is not mentioned in the review and should be incorporated.
I suggest adding a table with the main inflammation mediators involved in plaque formation, instead of referring the reader to bibliography (line 125).
I believe that the figures are not informative. The first one is not at cell or molecular level, while the second one is too general. I think that according to the title, a figure explaining the RhoA pathway and its implication on macrophage activation should be provided instead.
Minor comments:
The expression “bad cholesterol” should be avoided in a highly scientific work like this. (line 86)
LDLs are not nanoparticles, and they also contain free cholesterol in the outer layer (line 88-89)
The web site mentioned in line 143 doesn’t work.
There are no LDL droplets in circulation. There are lipoprotein particles. (line 145)
In lines 181 and 186 the word “plague” must be amended.
Author Response
Reviewer 2:
Comments and Suggestions for Authors
Kloc M et alter have reviewed the role of macrophages and RhoA pathway in the development of atherosclerosis.
This is a comprehensive review of plaque formation, from the Stary pathological stages to molecular basis.
I believe the authors have performed a great job putting all together, the available information on molecular bases of atherosclerotic plaque formation. I believe the content of the review encompasses the current scientific knowledge and it will be useful for readers.
There are some aspects that I would like to be better addressed:
The role of cholesterol, and particularly LDL-C as a causal factor should be clearly stated. Although it is mentioned across the paper, I feel the importance of cholesterol as endothelium or macrophage activator, or the role of cholesterol crystals inducing inflammasome activation, is somehow diluted among other factors (lines 93-94). I suggest highlighting its etiological role that has been recently reviewed in Boren J et al Eur Heart J, 21;41(24):2313-2330.
RESPONSE: as suggested we added the information about the role of LDL-C and cholesterol crystals and added several references
I believe that in the context of this review it would be more important to devote some room to the role of HDL than, for example describing all Stary stages. HDL is not mentioned while playing a crucial role on maintaining the artery wall cholesterol burden under control, through its interaction with macrophages.
I suggest including a paragraph on this issue.
RESPONSE: We think that Star’s stages are important for the readers less familiar with the subject, but as suggested we added the paragraph on the role of HDL and added the references
It seems that there are two different phenotypes of macrophages in arteriosclerotic plaques. M1 proinflammatory and acting as plaque disruptor and M2, which are anti-inflammatory and taking part of healing mechanisms (Vergallo R et al. NEJM 2020: 383, 9 and Eur Heart J 2020). This aspect is not mentioned in the review and should be incorporated.
RESPONSE: as suggested we added this subject to the review, and added references
I suggest adding a table with the main inflammation mediators involved in plaque formation, instead of referring the reader to bibliography (line 125).
RESPONSE: We believe that referring the reader to the tables in Tedgui and Mallat paper, which is an extremely comprehensive review containing several tables of different factors involved in atherosclerosis and plaque formation, is much better than making a table containing a partial information
I believe that the figures are not informative. The first one is not at cell or molecular level, while the second one is too general. I think that according to the title, a figure explaining the RhoA pathway and its implication on macrophage activation should be provided instead.
RESPONSE: we changed the Figure 1, and also added Figure 3, which is the extension of Figure 2, and also added Figure 4 that depicts the role of RhoA pathway in the macrophages/foam cells
Minor comments:
The expression “bad cholesterol” should be avoided in a highly scientific work like this. (line 86)
RESPONSE: we removed the term “bad cholesterol”
LDLs are not nanoparticles, and they also contain free cholesterol in the outer layer (line 88-89)
RESPONSE: as suggested we changed the description
The web site mentioned in line 143 doesn’t work.
RESPONSE: we checked the website and it works but has to be copied/pasted. We added this information after the website name
There are no LDL droplets in circulation. There are lipoprotein particles. (line 145)
RESPONSE: as suggested we corrected this statement
In lines 181 and 186 the word “plague” must be amended.
RESPONSE: we corrected this throughout the text

Round 2
Reviewer 1 Report
The authors are strongly recommended to check through typos and grammar.
- line 187, should be "Fig. 2 and Fig. 3"
- line 223-224
- line 238
- line 250
- line 290
- line 359
- line 453-454
- line 460-464
- line 468
Reviewer 2 Report
No further comments